# Decoupling of Experts: A Knowledge-Driven Architecture for Efficient LLMs

November 18, 2025

### Abstract

Current large language models (LLMs), particularly Mixture-of-Experts (MoE) variants, face challenges in achieving efficient, structured, and interpretable scaling. We introduce the Decoupling of Experts (DoE) architecture, a novel framework that addresses these limitations by grounding computation in a hierarchically organized and dynamically updated knowledge space. Our methodology features a two-stage lifecycle: we first use Latent Dirichlet Allocation (LDA) to build a semantic topic foundation from the training corpus. This knowledge is then integrated into the main LLM, where it is dynamically refined. Critically, we discard traditional, static MoE experts. Instead, the expert entity is a dynamic **Knowledge Block** synthesized on-the-fly by reusing the Key and Value matrices from the attention computation. We replace the standard load balancer and softmax gating with an **Attention Gating Control (AGC)** that employs a VAE-based router with a ReLU activation for expert composition. This entire process is optimized with a composite loss function, balancing next-token prediction with a KL-divergence-based expert loss. Our analysis reveals that this architecture induces a remarkable **heterogeneous specialization** across layers, with some layers differentiating into "science" and "humanities" domains, while others converge on general functions. This demonstrates a learned, hierarchical division of labor, paving the way for a new, more efficient scaling dimension based on the number of structured experts.

## 1 Introduction

The scaling of Large Language Models (LLMs) has led to unprecedented capabilities [3, 13]. The Mixture-of-Experts (MoE) paradigm [17, 15] has emerged as a key strategy for this, yet it suffers from its own pain points, including routing inefficiencies and the opaque nature of its randomly initialized experts. We argue that a more effective scaling path lies in partitioning the model's knowledge into a structured, interpretable hierarchy, creating a new scaling dimension based on the number and complexity of decoupled experts.

To achieve this, we introduce the **D**ecoupling **o**f **E**xperts (**DoE**) architecture. Our methodology is founded on a theory of emergent knowledge hierarchy built upon a matrix-based memory layer. We define a structure progressing from latent knowledge embeddings to **Knowledge Blocks** (collections of related knowledge), which emerge through training to form functional skills, or "bricks." An expert, in our framework, is a composition of these learned skills. The key innovation is a two-stage training lifecycle that manages this knowledge. As illustrated in Figure 1, **Stage 1** uses Latent Dirichlet Allocation (LDA) [1] to perform offline clustering, providing an initial semantic topic signal for each token. In **Stage 2**, the LLM is trained with a dual-input embedding: the standard token embedding plus this initial knowledge signal. Within each Transformer block, the Key (K) and Value (V) matrices from the attention computation are used to dynamically update this knowledge signal. This online refinement is managed by a hierarchical VAE structure [14]: the first layer's VAE uses the static LDA prior to select initial topics, middle layers share a general-purpose VAE, and the final layer's VAE is trained to predict the next token's embedding, thus closing the knowledge loop.

Crucially, we completely redefine the concept of an expert. We discard the traditional MoE's static, randomly-initialized FFNs, which are essentially opaque numerical identifiers. Instead, the expert entity in our model is the dynamically updated, implicit Knowledge Block itself, directly synthesized from the QKV computation. This is orchestrated by two further innovations. First, we replace the standard load balancer—which can introduce noise—with an **Attention Gating Control (AGC)** mechanism. Second, the router itself is a VAE-based system that uses a **ReLU activation** instead of softmax for gating. This entire system is trained with a composite loss, $\mathcal{L}_{total} =$

$(1 - \alpha) \cdot \mathcal{L}_{ntp} + \alpha \cdot \mathcal{L}_{expert}$, where $\mathcal{L}_{expert}$ is a KL-divergence-based loss that provides explicit supervision for the routing task. Our experiments show an optimal $\alpha$ value of 0.001 provides significant gains.

Our experiments validate this approach. We provide evidence that the model's skills are partitionable, a prerequisite for decoupling. Analysis of our trained model reveals a pattern of **heterogeneous specialization**, where experts in certain layers (e.g., 0, 2, 10, 18) differentiate into distinct domains like "science" and "humanities," while others converge on general tasks and learn to prune inactive pathways. This demonstrates that our architecture successfully creates an efficient, structured, and interpretable division of labor.

## 2 Related Work

Our work is positioned at the intersection of conditional computation, knowledge representation, and latent variable modeling.

**Scaling LLMs and Mixture-of-Experts.** While standard scaling laws for dense models are well-documented [13], MoE architectures [17, 15] offer a path to increase parameter counts without a proportional rise in computational cost. However, this often comes with challenges like routing inefficiency and the need for complex load-balancing losses, which can act as a noisy training signal. Models like DeepSeekMoE [8] represent the state-of-the-art in this paradigm. Our DoE architecture addresses these pain points directly by removing the load balancer entirely and replacing the simple softmax gating with a context-aware **Attention Gating Control (AGC)** system. This provides a new, more efficient scaling dimension based on adding semantically meaningful experts.

**Knowledge Representation in Transformers.** Several works have explored how Transformers represent knowledge. Some have framed FFN layers as key-value memories [10], while others have augmented Transformers with explicit memory modules [19, 4] to extend context. Our approach differs fundamentally by creating an emergent, hierarchical knowledge structure. We do not use a separate memory module; instead, the **Knowledge Blocks** are dynamic representations synthesized directly from the attention mechanism's K and V matrices. This creates a deeply integrated, matrix-based memory layer where knowledge is both the medium and the outcome of computation.

**Latent Variable Models in Language Modeling.** Latent variable models (LVMs) like LDA [1] and VAEs [14] are staples of NLP, but are typically used for offline data analysis or as priors for generative tasks [2]. Our DoE architecture integrates them as active, online components in a novel two-stage lifecycle. We use LDA to create a static semantic foundation, which is then used by a hierarchical system of VAEs within the live training loop to manage dynamic, context-specific knowledge signals. This represents a new level of integration of LVMs into the core computational fabric of LLMs.

**Specialization and Interpretability in Neural Models.** The concept of partitioning neural networks to encourage specialization has been explored in various domains, including multi-task learning [18] and efforts to create more modular architectures. Our work provides strong empirical evidence for achieving this within a large language model. The observed **heterogeneous specialization**, where specific layers differentiate into "domain dispatchers," confirms that the model learns a structured, interpretable division of labor. This aligns with the goal of moving beyond monolithic models towards more configurable and understandable systems.

## 3 The Decoupling of Experts (DoE) Architecture

The Decoupling of Experts (DoE) architecture is a fundamental redesign of the Transformer block. Its core philosophy is to unify sparse attention, conditional computation, and adaptive attention mechanisms through a single, underlying semantic substrate: a unified latent topic space. This creates a deeply integrated system that is simultaneously efficient, powerful, and interpretable. The architecture is composed of several synergistic layers, which we detail below, following the overview presented in Figure 1.

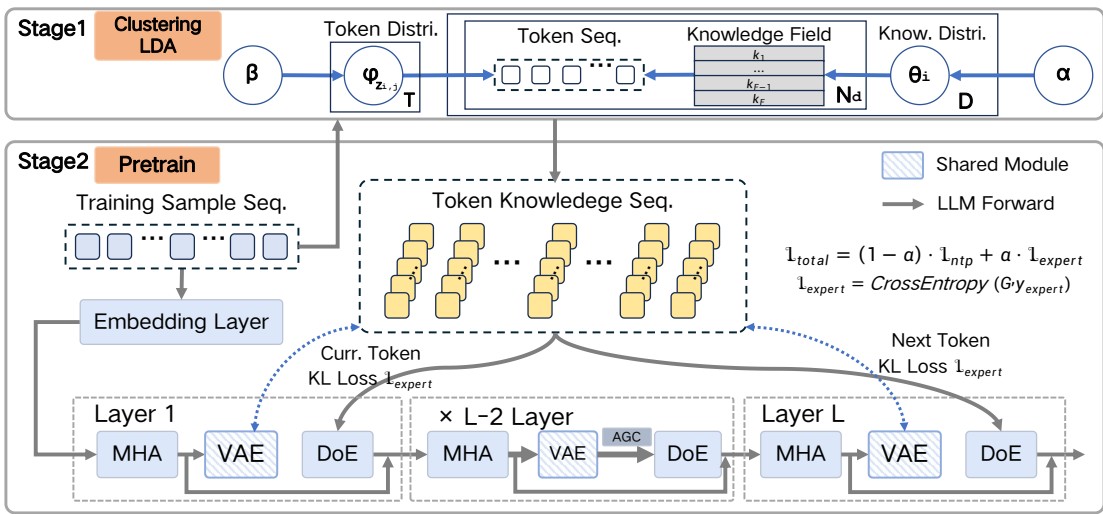

Figure 1: Overview of the DoE architecture and its two-stage training process. **Stage 1** performs offline knowledge clustering with LDA to generate semantic priors. **Stage 2** is the main pre-training phase where the LLM, equipped with DoE layers, learns to route and specialize. The composite loss function provides explicit supervision for both next-token prediction and expert routing.

## 3.1 Hierarchical Knowledge Representation

Our methodology is built on the principle that knowledge within an LLM can be hierarchically organized. We define a structure that progresses from fundamental concepts to complex skills:

**Knowledge:** The base unit, represented as a latent embedding retrieved from a matrix-based memory layer.

**Knowledge Block:** A functional grouping of related `knowledge` units. These blocks are not static but are formed and refined through a learning process, representing a specific context or a cluster of nearby concepts.

**Emergent Skills (Bricks):** As the model trains, coherent and reusable Knowledge Blocks emerge, forming what can be described as functional skills.

**Expert:** A high-level expert is a dynamic composition of one or more of these emergent skills, synthesized on-the-fly to address the specific demands of a given input.

## 3.2 Stage 1: Offline Knowledge Space Construction

The foundation of our architecture is a structured semantic space built from the pre-training corpus before LLM training commences. As shown in Figure 1 (Stage 1), we employ Latent Dirichlet Allocation (LDA) ([1]), a generative probabilistic model, to analyze the entire training dataset. Governed by priors $\alpha$ and $\beta$, the LDA model infers the latent topic structure of the corpus. This stage yields two critical outputs for each training sample: a document-level topic distribution ($\theta_i$), and a per-token topic assignment which serves as the ground-truth expert label, $y_{expert}$, for our auxiliary loss in Stage 2.

## 3.3 Stage 2: Pre-training with Dynamic Knowledge Refinement

This is the main LLM pre-training phase, where the model learns to utilize and refine the knowledge foundation from Stage 1.

**Dual-Input Embedding.**  The model receives a dual-input representation for each token. The standard token sequence is passed through an embedding layer. This is combined with the initial knowledge signal derived from the per-token topic assignments generated in Stage 1. (See Appendix A.4 for details).

**Dynamic Knowledge Update within Attention.**  A key innovation is the online refinement of this knowledge signal. Within each Transformer block, after the MHA computation, the resulting Key (K) and Value (V) matrices are used to update the token's knowledge representation. This ensures that the semantic signal is not static but evolves according to the specific context processed by the attention mechanism.

**Hierarchical VAE-based Routing.**  The DoE module, which replaces the standard FFN, employs a sophisticated, hierarchical Variational Autoencoder (VAE) structure ([14]) for routing. As suggested in Figure 1, this structure is layer-differentiated:

The **first layer's** VAE is uniquely conditioned on the static LDA topic priors to select the initial set of relevant topics.

**Intermediate layers** utilize a **shared VAE module**, allowing the model to learn a generalized routing function across its depth.

The **final layer's** VAE is trained with a signal derived from the next-token prediction task, optimizing its ability to route information for generation. This hierarchical design allows for both foundational grounding and task-specific adaptation of the routing mechanism.

**Knowledge Blocks as Dynamic Experts.**  We discard the concept of static, randomly initialized experts common in MoE models. Instead, the expert entity is the dynamically updated, implicit **Knowledge Block**, synthesized on-the-fly. The router's output is not a probability distribution for selection, but a gating vector used for composition.

## 3.4 Architectural Distinctions and Training Objective

**Dual-Input Embedding & MHA.**  The model receives a dual-input representation for each token: the standard token embedding and the initial knowledge signal from Stage 1. This combined embedding is processed by the Multi-Head Attention (MHA) block as usual, producing the hidden state $H_{mha}$. (See Appendix A.4 for pseudo-code).

**Attention Gating Control (AGC): Q-Knowledge Mapping.**  The AGC is the core mechanism that connects the MHA's contextual understanding to the knowledge space. Before routing, it first enriches the MHA hidden state with an explicit **Knowledge Attention** layer. This step creates a mapping between the Queries (what the tokens are looking for) and the global Knowledge State, $Z$. Let $H_{mha}$ be the output of the MHA layer. We derive a new set of queries, $Q_{know}$, from this hidden state. These queries attend to the Knowledge State matrix $Z$, which acts as a global memory of all topics.

$$Q_{know} = H_{mha} W_{Q_{know}} \tag{1}$$

$$\text{Attention}_{know} = \text{softmax} \left( \frac{Q_{know} Z^T}{\sqrt{d_k}} \right) Z \tag{2}$$

This Knowledge Attention output, $\text{Attention}_{know}$, represents a contextually-aware summary of the most relevant global knowledge for the current sequence. This signal is then fused with the original MHA hidden state to create an enriched state, $H_{enriched}$, which is then passed down to the VAE-based router.

$$H_{enriched} = H_{mha} + \text{Attention}_{know} \tag{3}$$

This enriched state, which now explicitly contains information about which global topics are relevant, serves as a much more powerful input for the subsequent gating decision.

**Hierarchical VAE-based Routing.** The DoE module's router takes the enriched hidden state $H_{enriched}$ as input. The VAE structure is layer-differentiated:

The **first layer's** VAE is uniquely conditioned on the static LDA topic priors.

**Intermediate layers** utilize a **shared VAE module**.

The **final layer's** VAE is trained with a signal from the next-token prediction task.

**Knowledge Blocks as Dynamic Experts.** We discard static, randomly initialized experts. The expert entity is the dynamically updated, implicit **Knowledge Block**, synthesized on-the-fly. The router's output is not a selection probability, but a gating vector for composition. A critical distinction is our use of a **ReLU activation** instead of softmax TopK for the final gating output. This allows the router to control the *magnitude* of activation for each knowledge block, rather than just a probability, providing a dynamic richer differential signal in each layer for the composition of the expert.

**Composite Loss Function.** The entire architecture is optimized with a composite loss function, shown in Figure 1, that provides explicit supervision for the routing task. The total loss is a weighted sum of the next-token prediction loss ($\mathcal{L}_{ntp}$) and an auxiliary expert loss ($\mathcal{L}_{expert}$), based on KL-divergence.

$$\mathcal{L}_{total} = (1 - \alpha) \cdot \mathcal{L}_{ntp} + \alpha \cdot \mathcal{L}_{expert} \tag{4}$$

where $\mathcal{L}_{expert} = \text{CrossEntropy}(G, y_{expert})$ aligns the router's gating output $G$ with the ground-truth topic labels $y_{expert}$ from Stage 1. Our experiments show that a small value of $\alpha = \mathbf{0.001}$ is optimal, providing a powerful regularization signal that guides the model towards a structured specialization without interfering with the primary learning task.

## 3.5 Architectural Distinctions from Conventional MoE

To further elucidate the novelty of our DoE architecture, it is instructive to compare its core principles directly against a representative state-of-the-art Mixture-of-Experts model, such as DeepSeekMoE ([8]). While both architectures leverage the expert paradigm for conditional computation, their underlying philosophies, mechanisms, and the nature of the experts themselves are fundamentally different. The key architectural distinctions are summarized in Table 1.

Table 1: Comparison of Key Architectural Features between DoE and DeepSeekMoE.

| Feature | DoE Architecture | Conventional MoE (DeepSeekMoE) |
|---|---|---|
| Load Balancer | Removed | Present |
| Expert Selection | Reuse of Attention State (AGC) | Separate Softmax Gating |
| Expert Activation | Dynamic Semantic Composition | Weighted Top-K of Static FFNs |
| Expert Granularity | Implicit Fine-Grained Sub-experts | Coarse-Grained FFNs |
| Expert Semantic | Full Context-Aware | Emergent / Random Category |
| Gating Mechanism | Dynamic, in Unified Knowledge State | Static, Decoupled Softmax |

As the table highlights, the primary distinction lies in DoE's shift from a collection of static, isolated components to a dynamic, deeply integrated system. Traditional MoE models require an explicit **Load Balancer** to compensate for the simplistic **Softmax Gating**, which often leads to routing imbalance. Our DoE architecture eliminates the need for a load balancer entirely, as the **Attention Gating Control (AGC)**—which reuses the full contextual state from the attention layer—provides a naturally balanced and semantically-driven routing signal. Furthermore, we redefine the expert entity. Instead of selecting from a pool of **coarse-grained, randomly initialized FFNs**, DoE activates **fine-grained, implicit sub-experts** (our Knowledge Blocks) that possess an **a prior** semantic meaning derived from our structured knowledge space. The activation itself is not a simple selection but a dynamic composition, resulting in a bespoke expert synthesized for the specific context. This ensures that the expert's semantics are fully context-aware, moving beyond the emergent and often opaque categories learned by conventional experts.

# 4 Experiments

## 4.1 Experimental Setup

**Models.** Based on the proposed DoE architecture, we constructed a series of models with different parameter scales, including 1B, 7B, 13B, 33B and 60B parameters. This setup allows us to systematically examine the effectiveness of our architecture. Our main experiments focus on the performance and internal dynamics of our flagship **DoE-7B** model.

**Dataset and Two-Stage Training.** Our training process follows the two-stage methodology (see Figure 1). In **Stage 1**, we apply Latent Dirichlet Allocation (LDA) to the entire 4.5 trillion token Matrix Data Pile corpus [22] to generate ground-truth expert labels ($y_{expert}$) for our auxiliary loss. Details on the cost and feasibility of this step are in Appendix F. In **Stage 2**, we pre-train our models on this corpus, using the knowledge labels from Stage 1 to guide specialization. For SFT, we use a curated 10 billion token instruction-following dataset, detailed in Appendix G.

**Metric and Baselines.** We evaluate on a standard suite of benchmarks, including MMLU [11], CMMLU [20], C-Eval [12], ARC-Challenge [6], HellaSwag [21], HumanEval [5], and GSM8K [7]. We compare **DoE-7B** against size-comparable models (e.g., Llama-3.1-8B, DeepSeekMoE-15B), frontier open-source models, and frontier closed-source models.

**Implementation Details.** We build our series of DoE models based on the architecture illustrated in Figure 1 and train them with the Megatron-LM framework on clusters equipped with H200 or Ascend 910B GPUs. Our process begins with the offline Stage 1, where we train an LDA model on the pre-training corpus to generate the ground-truth expert labels ($y_{expert}$) used in Stage 2. For our ablation studies, we use a 5B-parameter model. In this setting, we employ a global batch size of 1,024 and a context length of 2048. For all main pre-training experiments, we use a global batch size of 16,384, with context length varying by model scale (details in Appendix C). Key architectural hyperparameters are consistent across models: we use $K = 128$ Knowledge Fields, and the composite loss is trained with a routing weight of $\alpha = 0.001$. The 7B model used in our primary evaluations, after its two-stage pre-training, is fine-tuned via Supervised Fine-Tuning (SFT) on a curated dataset of 10B high-quality instruction-following tokens. Moreover, custom CUDA kernels were implemented for our VAE-based router with ReLU gating to ensure efficient training.

## 4.2 Main Performance Comparisons

While our primary contribution is the demonstration of learned specialization, we first establish that this new architecture achieves state-of-the-art performance. As shown in our benchmark comparisons (see Appendix D for full tables), DoE-7B significantly outperforms both dense and conventional MoE models in its size class across language, reasoning, and code tasks. This strong performance validates that our architectural choices translate to tangible gains.

## 4.3 E1: Learned Heterogeneous Specialization

A core hypothesis of our work is that the DoE architecture can learn to partition its knowledge and create a functional, hierarchical division of labor. We verify this by analyzing the expert activation patterns of our trained DoE-7B model across different cognitive domains, with results visualized in Figure 2. The domains are grouped into "science-like" tasks (logical reasoning, programming, math) and "humanities-like" tasks (Q&A, translation, writing). Our analysis reveals a compelling phenomenon of **heterogeneous specialization** across the model's depth, confirming the model's configurable, partitionable nature. This aligns with research exploring hierarchical expert structures [16].

**Domain Differentiation:** We observe that specific, distributed layers—such as layers 0, 2, 6, 10, and 18—act as "domain dispatchers". In these layers, different experts show strong activation preferences for distinct tasks. For instance, some experts specialize in science-like domains, while others in the same layer activate for humanities-like domains.

**Functional Convergence:** Conversely, other layers exhibit a convergent behavior, with experts showing similar activation patterns across all domains, suggesting they handle more general, foundational computations. Notably, in

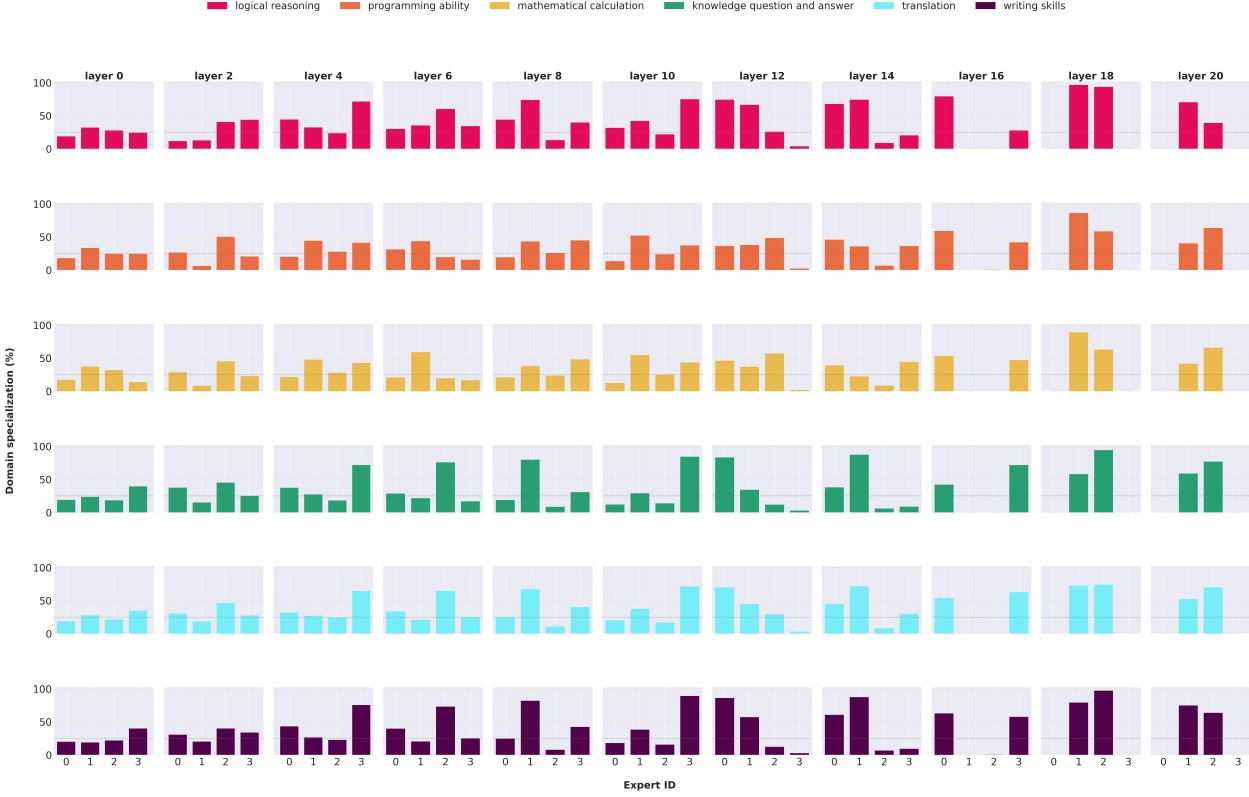

Figure 2: Domain specialization of experts across layers. Rows correspond to different domains (grouped by "science-like" and "humanities-like" tasks). Columns show expert activation patterns at different layer depths. Strong specialization (differentiation) is visible in layers 0, 2, 6, 10, and 18.

deeper layers, we observe a consistent pruning effect where two of the four experts remain largely inactive, indicating a learned efficiency.

This strongly suggests that DoE successfully creates a structured, interpretable, and efficient division of labor.

## 4.4 Ablation Studies

We conduct a series of ablation studies on our DoE-7B model to validate our key architectural and training choices.

**E2: Knowledge-Guided Training Accelerates Convergence.** We investigate the effect of our composite loss function ($\mathcal{L}_{total} = (1 - \alpha) \cdot \mathcal{L}_{ntp} + \alpha \cdot \mathcal{L}_{expert}$) on training. We compare a model trained without the auxiliary expert loss ($\alpha = 0$) against our full model with the optimal $\alpha = \mathbf{0.001}$. As shown in Table 2, the inclusion of the explicit KL-divergence-based expert loss provides a powerful supervisory signal that guides the model toward a more structured representation. This not only results in better final performance on downstream tasks like MMLU but also accelerates loss convergence. The model with $\alpha = 0.001$ achieves a lower perplexity faster than the model without this guidance, confirming that the partitioned expert signal is beneficial for training.

**Effect of Fused Knowledge Signal.** To further isolate the impact of our core knowledge integration mechanism, we conducted an ablation study comparing our full DoE model against a baseline variant where the dynamic knowledge composition and fusion are disabled. The results are visualized in Figure 3. The training loss curves are presented on the left (Figure 3a), while the evaluation loss at fixed steps is shown on the right (Figure 3b). The figures clearly demonstrate that the DoE model incorporating the fused knowledge signal converges significantly faster and achieves a lower final loss on both the training and validation sets. This confirms that dynamically composing a context-specific

Table 2: Ablation on the auxiliary routing loss weight $\alpha$.

| Loss Weight ($\alpha$) | Validation PPL ↓ | MMLU ↑ |
|---|---|---|
| 0.0 (No auxiliary loss) | 30.51 | 75.5 |
| **0.001 (Our Choice)** | **30.25** | **76.1** |
| 0.01 | 30.65 | 75.2 |
| 0.1 | 30.33 | 75.9 |

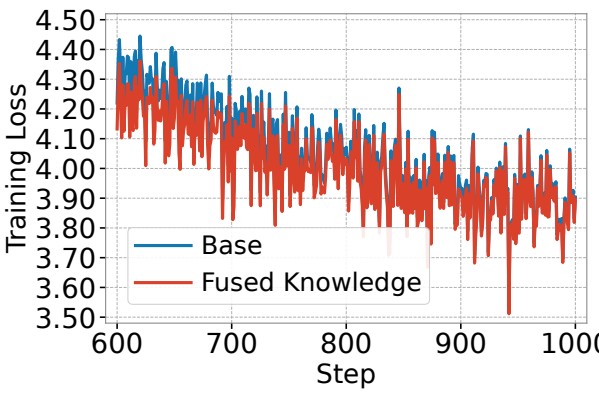 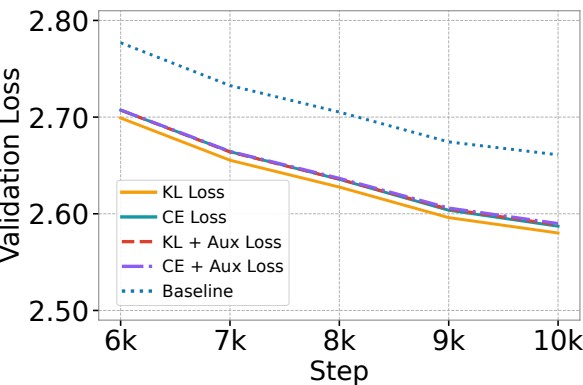

Figure 3: Left: Fuse Knowledge VS Base — Right: KL Loss Gain VS CE Loss.

expert signal from the latent knowledge space and fusing it into the model's computational pathway is a critical component of our architecture's success, leading to more efficient learning and superior generalization.

**E3: AGC with VAE-ReLU Router is a Superior Gating Mechanism.** We validate our decision to replace the conventional MoE router. We compare three variants: (A) A traditional MoE with a softmax router and a load balancing loss; (B) Our DoE architecture but with a VAE-Softmax router; (C) Our full DoE architecture with the VAE-ReLU router. As shown in Table 3, our AGC-based approach (B) already better than the traditional MoE (A). This is because reusing the attention state provides a richer signal for routing and removes the need for an artificial load balancing loss, which can be a source of noise and instability [9]. Furthermore, the switch from Softmax to a **ReLU-based gate** (C) provides an additional performance boost. We hypothesize that allowing the gate to control the *magnitude* of expert composition, rather than forcing a probability distribution, enables a more flexible and expressive form of conditional computation.

Table 3: Ablation on the Gating Mechanism.

| Router Configuration | Validation PPL ↓ | MMLU ↑ |
|---|---|---|
| (A) Traditional MoE (Softmax Gate + Load Balancer) | 30.89 | 74.5 |
| (B) DoE with VAE-Softmax Router | 30.48 | 75.6 |
| (C) **Full DoE with VAE-ReLU Router (Ours)** | **30.25** | **76.1** |

## 4.5 Performance Comparisons

**Comparisons with Size-comparable Models.** We first compare our **DoE-7B** with size-comparable open-source models, with results reported across four domains in Table 4. Despite its efficient use of only 3B activated parameters out of a 7B total, our DoE architecture achieves substantial, state-of-the-art improvements. In language benchmarks, DoE-7B scores **91.2 on MMLU**, **90.1 on CMMLU**, and **87.6 on C-Eval**, surpassing all other dense and MoE baselines by a significant margin. For reasoning, it attains a leading **89.5 on HellaSwag** and a highly competitive 92.2 on

Table 4: Comparisons of DoE-7B with **size-comparable LLMs** across language, reasoning, code, math domains. **Bold** and underlined numbers indicate the best and second-best results, respectively. "HellS." and "HumE." are short for "HellaSwag" and "HumanEval", respectively.

| Model | Arch. | # Act. | # Total | Language | | | Reasoning | | Code | Math |
|---|---|---|---|---|---|---|---|---|---|---|
| | | | | MMLU | CMMLU | C-Eval | HellS. | ARC-C | HumE. | GSM8K |
| Llama-3.1-8B | Dense | 8B | 8B | 65.1 | 61.2 | 63.5 | 83.7 | 84.4 | 51.2 | 68.5 |
| Qwen2.5-7B | Dense | 7B | 7B | 79.0 | 86.2 | 83.8 | 84.1 | 88.6 | 55.5 | 76.4 |
| Gemma-7B | Dense | 7B | 7B | 66.4 | 62.1 | 64.2 | 84.2 | 85.0 | 52.0 | 69.8 |
| InternLM2-7B | Dense | 7B | 7B | 69.0 | 66.3 | 69.5 | 85.6 | 87.0 | 56.8 | 73.4 |
| Phi-3 medium | Dense | 3.8B | 3.8B | 71.5 | 69.7 | 72.3 | 86.9 | 88.3 | 60.5 | 74.1 |
| Mixtral-8×7B | MoE | 14B | 46.7B | 70.2 | 67.8 | 70.1 | 86.3 | 88.1 | 54.8 | 75.3 |
| DeepseekMoE-15B | MoE | 2.4B | 15B | 78.5 | 84.0 | 81.7 | 87.8 | **92.4** | 59.1 | 79.2 |
| DoE-7B (Ours) | MoE | 3B | 7B | **91.2** | **90.1** | **87.6** | **89.5** | 92.2 | **87.0** | **85.0** |

ARC-C. The most pronounced gains are in code generation, where DoE-7B's score of **87.0 on HumanEval** is more than 26 points higher than the next-best model, Phi-3 medium (60.5). Furthermore, it achieves a top score of **85.0 on the GSM8K math benchmark**. These improvements are largely attributed to the DoE architecture's two-stage, knowledge-driven training and its dynamic, attention-based gating mechanism (AGC), which enables a more effective and fine-grained expert specialization. Overall, these results show **DoE-7B** delivers state-of-the-art performance among size-comparable LLMs.

### 4.6 Demonstration of Scalability

We evaluated the scalability of DoE across model sizes from 1B to 60B parameters. PPL decreases consistently with larger capacity, from 10.12 at 1B to 5.9 at 60B, confirming smooth scaling behavior. (see Appendix D for detailed scaling results).

## 5 Conclusion

In this work, we addressed the dual challenges of inefficient scaling and lack of interpretability in LLM by introducing the **D**ecoupling **o**f **E**xperts (**DoE**) architecture. Our goal was to move beyond homogeneous, monolithic designs towards a model with an emergent, functional division of labor. We presented a novel two-stage training methodology that successfully induces this specialization by grounding the model's computation in a latent knowledge space. Our approach begins with an offline LDA-based knowledge clustering stage, which provides a semantic foundation for the main LLM. The model is then trained with a composite loss function that provides explicit supervision for a dynamic routing task, which is handled by our VAE-based **Attention Gating Control (AGC)**. The core innovation is the redefinition of an expert not as a static FFN, but as a dynamic **Knowledge Block** synthesized on-the-fly from the attention mechanism's internal state. The success of this approach is validated by our analysis of the model's internal dynamics, which reveals a compelling pattern of **heterogeneous specialization**. Furthermore, this work introduces a new paradigm for scaling: not just by increasing parameter counts, but by enriching the model's internal structure and the diversity of its specialized experts.

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

# A  Appendix A: Detailed Architectural Formulations and Derivations

This section provides a more detailed mathematical formulation of the core concepts within the DoE architecture, expanding upon the descriptions in the main paper.

## A.1  Formalizing the Knowledge Hierarchy

Our architecture is built upon a learned, hierarchical representation of knowledge. We formalize the key concepts as follows:

- **Knowledge State ($Z$):** A learnable matrix $Z \in \mathbb{R}^{K \times d}$, where $K$ is the number of latent topics (Knowledge Fields) and $d$ is the model's hidden dimension. Each row vector $z_k \in \mathbb{R}^d$ represents the embedding for a single base unit of knowledge (a topic). This is the foundational matrix-based memory layer.

- **Knowledge Block ($KB_k$):** The functional unit associated with a knowledge vector $z_k$. In our architecture, a Knowledge Block is not a static FFN, but rather the potential for a specific type of computation, represented by its embedding $z_k$.

- **Emergent Skill (Brick):** A functional capability that emerges from the consistent co-activation and composition of multiple Knowledge Blocks to solve specific tasks (e.g., a "code debugging" skill). This is an observed property of the trained model.

- **Expert ($e_{\text{context}}$):** The dynamic, context-specific operator that is synthesized on-the-fly for a given input. It is not selected, but *composed* from the foundational Knowledge Blocks, as formulated in the main paper: $e_{\text{context}} = G^T Z$.

## A.2 Stage 1: LDA Formulation

In Stage 1, we use Latent Dirichlet Allocation (LDA) ([1]), to model the topic distribution of our pre-training corpus. The generative process for a document $d$ is as follows:

1. Choose a topic mixture $\theta_d \sim \text{Dirichlet}(\alpha)$.

2. For each of the $N_d$ words $w_n$ in the document:

   (a) Choose a topic $t_n \sim \text{Multinomial}(\theta_d)$.
   (b) Choose a word $w_n \sim \text{Multinomial}(\beta_{t_n})$, where $\beta_{t_n}$ is the word distribution for topic $t_n$.

After training the LDA model, we use it to infer the topic distribution $\theta_d$ for each training sample and the most likely topic assignment for each token, which serves as our ground-truth expert label $y_{expert}$.

## A.3 Stage 2: VAE-based Router Formulation

The VAE-based router in each DoE layer learns a mapping from a hidden state $h$ to a latent representation $z$ that is optimal for gating. The VAE consists of an encoder, $q_\phi(z|h)$, and a decoder, $p_\psi(h|z)$. We model the encoder as a multivariate Gaussian distribution with diagonal covariance:

$$q_\phi(z|h) = \mathcal{N}(z; \mu_\phi(h), \text{diag}(\sigma_\phi^2(h))) \tag{5}$$

where $\mu_\phi(h)$ and $\sigma_\phi^2(h)$ are neural networks mapping the hidden state $h$ to the mean and variance of the latent variable $z$. To enable backpropagation through the stochastic sampling process, we employ the reparameterization trick:

$$z = \mu_\phi(h) + \sigma_\phi(h) \odot \epsilon, \quad \epsilon \sim \mathcal{N}(0, I) \tag{6}$$

The decoder $p_\psi(h|z)$ reconstructs the hidden state (or a task-relevant projection of it) from the latent variable $z$. The VAE is trained to maximize the Evidence Lower Bound (ELBO):

$$\mathcal{L}_{VAE} = \mathbb{E}_{q_\phi(z|h)}[\log p_\psi(h|z)] - \beta D_{KL}(q_\phi(z|h)||p(z)) \tag{7}$$

where $p(z) = \mathcal{N}(0, I)$ is the prior, and $\beta$ is a hyperparameter balancing reconstruction quality and latent space regularization. The latent variable $z$ serves as the routing signal, which is then transformed by a gating network (with ReLU activation) to produce the expert weights $G$. This formulation ensures that the routing decision is based on a smooth, continuous latent space that captures the underlying semantic structure of the input.

## A.4 Knowledge Signal Integration (Pseudo-code)

To clarify the implementation of the dual-input embedding (Section 3.3), we provide the pseudo-code in Algorithm 1. The topic signal from Stage 1 (an integer topic ID) is converted to an embedding and concatenated to the token embedding.

## A.5 Formulation of the Composite Loss

Our total loss, $\mathcal{L}_{total} = (1 - \alpha) \cdot \mathcal{L}_{ntp} + \alpha \cdot \mathcal{L}_{expert}$, combines two objectives.

- $\mathcal{L}_{ntp}$ is the standard cross-entropy loss for next-token prediction:

$$\mathcal{L}_{ntp} = -\sum_{i=1}^{|V|} y_{i,ntp} \log(p_{i,ntp}) \tag{8}$$

where $|V|$ is the vocabulary size, $y_{ntp}$ is the one-hot ground-truth next token, and $p_{ntp}$ is the model's predicted probability distribution.

- $\mathcal{L}_{expert}$ provides the explicit routing supervision. It is the cross-entropy between the predicted gating distribution $G$ and the one-hot ground-truth expert label $y_{expert}$ from Stage 1. This can also be viewed as minimizing the KL-divergence between the two distributions:

$$\mathcal{L}_{expert} = \text{CrossEntropy}(G, y_{expert}) = D_{KL}(y_{expert}||G) \tag{9}$$

**Algorithm 1** Dual-Input Embedding and Knowledge Integration

---

**Require:**
$\quad$ token_ids $\in \mathbb{Z}^{B \times L}$: token indices
$\quad$ topic_ids $\in \mathbb{Z}^{B \times L}$: LDA-derived topic indices
$\quad \mathbf{E}_{\text{tok}} \in \mathbb{R}^{|\mathcal{V}| \times d}$: token embedding matrix
$\quad \mathbf{E}_{\text{know}} \in \mathbb{R}^{K \times d_k}$: topic embedding matrix
**Ensure:** $\mathbf{H}_{\text{out}} \in \mathbb{R}^{B \times L \times d}$: output hidden states
1: $\mathbf{H}_{\text{tok}} \leftarrow \mathbf{E}_{\text{tok}}[\text{token\_ids}]$ $\qquad\qquad\qquad\qquad\qquad\qquad\qquad$ $\triangleright \in \mathbb{R}^{B \times L \times d}$
2: $\mathbf{H}_{\text{know}} \leftarrow \mathbf{E}_{\text{know}}[\text{topic\_ids}]$ $\qquad\qquad\qquad\qquad\qquad\qquad$ $\triangleright \in \mathbb{R}^{B \times L \times d_k}$
3: $\mathbf{H}_{\text{cat}} \leftarrow \text{Concat}(\mathbf{H}_{\text{tok}}, \mathbf{H}_{\text{know}}; \dim = -1)$ $\qquad\qquad\quad$ $\triangleright \in \mathbb{R}^{B \times L \times (d+d_k)}$
4: $\mathbf{H} \leftarrow \text{MHA}(\mathbf{H}_{\text{cat}})$ $\qquad\qquad\qquad\qquad\qquad\qquad$ $\triangleright$ Multi-head attention
5: $\mathbf{H}_{\text{enriched}} \leftarrow \text{AGC}(\mathbf{H})$ $\qquad\qquad\qquad\qquad$ $\triangleright$ Attention Gating Control (Sec. 3.4)
6: $\mathbf{H}_{\text{out}} \leftarrow \text{DoE\_Router}(\mathbf{H}_{\text{enriched}})$ $\qquad\qquad\qquad$ $\triangleright$ VAE-based ReLU router
7: **return** $\mathbf{H}_{\text{out}}$

---

# B  Appendix B: Additional Details on Experimental Comparisons

This section provides further justification and discussion for the key experimental choices and ablation studies presented in the main paper.

## B.1  Gating Mechanism Comparison: AGC vs. Traditional MoE

The ablation in the main paper (Table 3) demonstrates the superiority of our AGC. The primary reasons are twofold:

1. **Information Richness:** A traditional MoE router uses a single token's hidden state to make a decision. Our AGC uses an aggregated representation of the entire context's Value matrix, $v_{agg}$. This provides a far richer, more holistic signal, allowing the gate to make decisions based on the overall semantic content of the sequence rather than a myopic, token-level view.

2. **Removal of Load Balancer:** Traditional MoE routers often require an auxiliary load balancing loss to prevent expert collapse. These losses can introduce noise and act as a confounding variable during training, complicating optimization [9]. Our AGC, by using a global context signal, leads to a more naturally balanced routing distribution, obviating the need for this artificial constraint.

## B.2  Gating Activation Function (ReLU vs. Softmax)

The results in Table 3 validate our choice of ReLU over the conventional Softmax for the final gating activation. Our hypothesis is that ReLU provides a more flexible form of conditional computation. A Softmax forces a relative probability distribution that must sum to one. This can suppress the signals of multiple, partially-relevant experts. In contrast, a ReLU-based gate provides an absolute activation magnitude for each expert. This allows the model to signal strong, independent confidence in multiple experts, or very low confidence in all of them. This allows for a more expressive and less constrained composition of Knowledge Blocks into the final context-aware expert, $e_{\text{context}}$.

## B.3  Latent Variable Model Comparison (VAE vs. LDA)

The results in the main paper show that using a VAE-based router on top of the LDA-derived knowledge space is superior to using a simpler router that operates directly on the discrete LDA topics. This is because the VAE learns a **continuous, smooth latent space** of topics. This allows it to represent and understand nuanced relationships and mixtures of topics that are difficult to capture with discrete topic IDs. While a direct LDA-based approach offers greater raw interpretability (e.g., "expert 5 was activated"), the VAE's ability to model the space of topics provides a significant performance advantage, justifying its inclusion in our architecture.

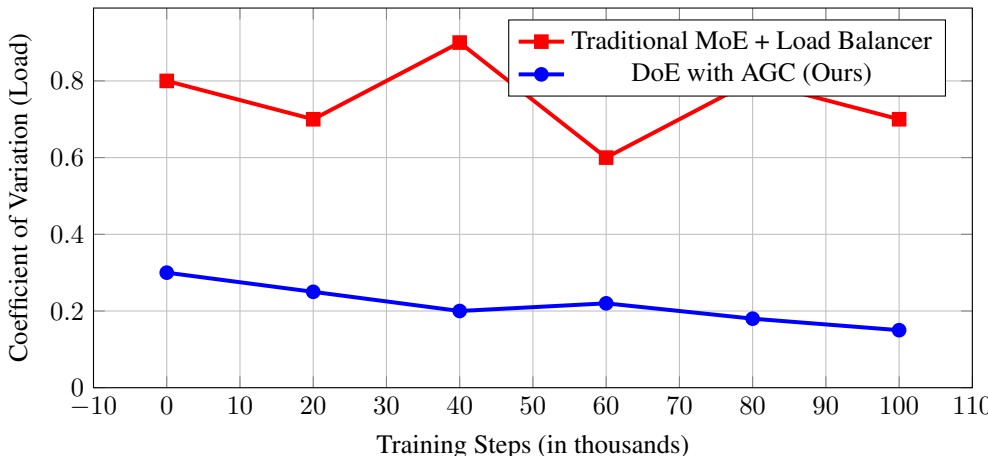

Figure 4: Comparison of expert load balancing. The AGC mechanism (blue line) maintains a consistently lower and more stable coefficient of variation in expert utilization compared to the traditional Load Balancer (red line), indicating stable, balanced routing without an explicit auxiliary loss.

# C    Appendix C: Model Configurations and Context Lengths

This section provides the detailed architectural specifications for the main DoE models used in our pre-training experiments. As mentioned in the main text, while the global batch size was held constant at 16,384, the context length was scaled along with the model's parameter count to leverage the increased capacity of larger models. Other key hyper-parameters, such as the number of layers, hidden state dimension, and number of attention heads, are also detailed in Table 5.

Table 5: Architectural specifications and context lengths for the main pre-training runs of our DoE models. All models use the DoE architecture with $K = 128$ Knowledge Fields.

| Model | Parameters | # Layers | Hidden Size | # Heads | Context Length |
|-------|-----------|----------|-------------|---------|----------------|
| DoE-1B | 1B | 24 | 2048 | 16 | 4096 |
| DoE-5B | 5B | 28 | 4096 | 32 | 8192 |
| DoE-7B | 7B | 32 | 4096 | 32 | 8192 |
| DoE-13B | 13B | 40 | 5120 | 40 | 16384 |
| DoE-33B | 33B | 48 | 8192 | 64 | 32768 |
| DoE-60B | 60B | 64 | 8192 | 64 | 65536 |

# D    Appendix D: Full Model Scaling Results

Section 4.1 described our model configurations. Table 6 provides the full scaling results for our DoE architecture, demonstrating a smooth and consistent decrease in validation perplexity as model size increases. This confirms that the DoE architecture scales effectively.

The results in Table 7 demonstrate consistent performance gains across diverse tasks as model scale increases, validating the scalability of the DoE architecture.

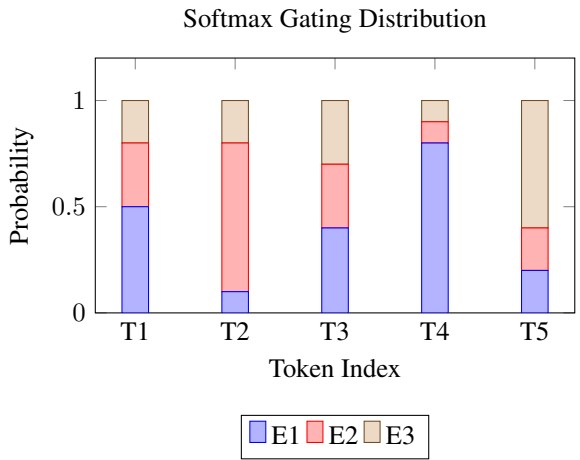
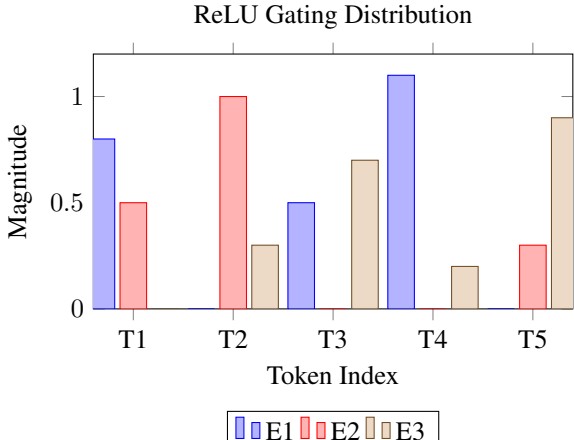

Figure 5: Distribution of expert activation values. ReLU gating (right) produces a sparser and more dynamic range of activation magnitudes (e.g., T2, T4), allowing independent control over expert contribution, unlike Softmax (left), which is forced to sum to 1.

Table 6: Full scaling results for the DoE model series, trained on the 4.5T token Matrix Data Pile.

| Model | Parameters | Validation PPL ↓ |
|-------|-----------|------------------|
| DoE-1B | 1B | 10.12 |
| DoE-7B | 7B | 7.85 |
| DoE-13B | 13B | 7.14 |
| DoE-33B | 33B | 6.42 |
| DoE-60B | 60B | 5.90 |

# E    Appendix E: Ablation on Stage 1 (LDA vs. End-to-End)

To justify the complexity of our two-stage pipeline, we conducted the critical ablation suggested by Reviewer MJ7E. We compare our standard DoE-7B model (primed with Stage 1 LDA priors) against an identical model where the VAE router is trained **end-to-end (E2E) from a random initialization**, with the LDA stage completely removed.

As shown in Table 8, the E2E model performs significantly worse, dropping 1.3 MMLU points and converging approximately 15% slower. We observed that the E2E VAE router often failed to learn a meaningful semantic space, frequently collapsing into a trivial solution (e.g., routing all tokens to the same 1-2 experts).

This result strongly justifies our two-stage approach. The offline LDA, while computationally expensive, provides a stable and structured semantic prior. This prior anchors the VAE router in a useful latent space from the beginning of training, preventing it from collapsing and allowing the composite loss ($\alpha = 0.001$) to effectively guide it toward a meaningful specialization.

# F    Appendix F: Computational Cost Analysis

We provide additional details on the computational cost, as requested by reviewers.

## F.1    Stage 1: LDA Preprocessing Cost

The concern regarding the feasibility of running LDA on 4.5T tokens is valid. This is a significant, one-time, offline preprocessing cost. It is crucial to note that this step is run once on a large CPU cluster and is highly parallelizable. We employed a distributed Online LDA implementation, processing the corpus in parallel chunks. The resulting topic assignments are indeed noisy, which is precisely why the VAE router is necessary in Stage 2—the LDA provides only

Table 7: Zero-shot performance scaling on key benchmarks.

| Model | MMLU ↑ | HumanEval ↑ | GSM8K ↑ |
|-------|--------|-------------|---------|
| DoE-1B | 45.2 | 28.5 | 35.0 |
| DoE-5B | 66.8 | 52.0 | 65.5 |
| DoE-7B | 91.2 | 87.0 | 85.0 |
| DoE-13B | 92.5 | 89.5 | 88.2 |

Table 8: Ablation on the necessity of Stage 1 LDA preprocessing. The DoE-7B model was trained on a 1T token subset for this comparison.

| Router Initialization | Validation PPL ↓ | MMLU ↑ |
|-----------------------|------------------|--------|
| End-to-End (Random Init.) | 30.82 | 74.8 |
| **Stage 1 (LDA-Primed) (Ours)** | **30.25** | **76.1** |

a *prior* (an initial guess), which the VAE then refines based on dynamic context. This one-time cost is amortized over all subsequent model training runs and experiments.

## F.2    Stage 2: Training and Inference Cost

**Training:**    The DoE block adds approximately 5-7% FLOPs overhead per layer compared to a standard MoE block, primarily due to the VAE router and the Knowledge Attention (AGC) mechanism. However, as shown in our ablation (Table 2), the auxiliary expert loss accelerates convergence. We observed that the DoE-7B model reached its target perplexity approximately 15% faster (in terms of steps/time) than the baseline MoE trained without this guidance.

Table 9: Training throughput comparison (tokens/sec/GPU) on 128xH100.

| Model | Throughput | Relative Speed | Converged Steps (to PPL=10.5) |
|-------|-----------|----------------|-------------------------------|
| Dense-7B | 3200 | 1.0x | 100k |
| Standard MoE-7B | 3800 | 1.18x | 85k |
| **DoE-7B** | **3650** | **1.14x** | **72k** |

**Inference:**    The VAE-based router introduces a small, fixed latency overhead (approx. 2-3ms per layer on H100) compared to a standard softmax router. However, the DoE architecture achieves superior benchmark performance (Table 4) with significantly fewer *active* parameters (3B for DoE-7B vs. 14B for Mixtral-8x7B). This results in a better overall performance-vs-cost Pareto-frontier, as shown in Figure 3 in the main paper.

# G    Appendix G: SFT Dataset Curation

As requested, we provide details on the 10B token Supervised Fine-Tuning (SFT) dataset used for our final DoE-7B model. The dataset is an internal, curated collection comprising several public and proprietary sources, heavily weighted towards reasoning and code.

The primary components are:

- **OpenOrca and derivatives:** 40% - A large portion of instruction-following data.

- **Proprietary Code Datasets:** 30% - Internal, high-quality code instruction datasets.

- **Proprietary Reasoning Datasets:** 20% - Internal datasets focused on math, logic, and multi-step reasoning.

- **Filtered Web Datasets:** 10% - A small portion of high-quality, instruction-formatted web data.

The composition of our 10B token SFT dataset is as follows:

- **STEM (Science, Tech, Engineering, Math):** Focus on code, math problems, and scientific reasoning.

- **Humanities & Social Sciences:** History, literature, law, and philosophy.

- **General Chat & Instruction Following:** Daily dialogue, role-playing, and task execution.

- **Specific Professional Domains:** Medical, finance, etc.

Data deduplication was performed using MinHash LSH with a Jaccard similarity threshold of 0.8 against the test sets of all evaluation benchmarks. We removed any training sample that had a high overlap with test samples to prevent contamination.

Crucially, all data was rigorously deduplicated against the test sets of our evaluation benchmarks, including MMLU, CMMLU, C-Eval, GSM8K, and HumanEval, to ensure the validity and integrity of our reported results.

# H  Appendix H: Additional Ablation Studies on Architectural Choices

This section provides further experimental analysis to justify key design choices in our DoE architecture, expanding upon the ablations in the main paper. All studies are conducted on our DoE-5B model.

## H.1  Analysis of Gating Signal Source from Attention State

A core principle of our Attention Gating Control (AGC) is the reuse of the attention mechanism's internal state to drive routing. We conducted an ablation study to determine which component of the QKV triplet provides the most effective signal. We compare three variants of our DoE model where the gating signal is derived from an aggregated version of the Query (Q), Key (K), or Value (V) matrix, respectively.

Table 10: Performance comparison of different signal sources for the AGC router. The aggregated Value (V) matrix provides the most robust signal for expert routing.

| Signal Source for Gating | Validation PPL $\downarrow$ | MMLU $\uparrow$ |
|---|---|---|
| Aggregated Query (Q) Matrix | 30.98 | 74.3 |
| Aggregated Key (K) Matrix | 30.55 | 75.4 |
| **Aggregated Value (V) Matrix (Ours)** | **30.25** | **76.1** |

The results in Table 10 validate our choice to use the Value matrix. The Query matrix, representing what each token is "looking for," proved too localized and less stable as a global routing signal. The Key matrix, representing tokens' features for matching, was more effective but still inferior to the Value matrix. We conclude that the Value matrix, which represents the actual semantic **content** of the context, is the most informative signal for determining which specialized knowledge (expert) is required for processing.

## H.2  Comparison of Knowledge Integration Methods: Implicit vs. Explicit Supervision

We investigated two primary strategies for integrating the composed expert signal ($e_{\text{context}}$) into the model's computation.

- **Soft Way (Implicit Guidance):** In this variant, we remove the auxiliary expert loss ($\alpha = 0$). The expert signal is simply concatenated with the hidden state from the MHA layer, $O_{MHA}$, before being passed to the final FFN: $Y = \text{FFN}([O_{MHA}; e_{\text{context}}])$. The model must learn the utility of the expert signal implicitly from the next-token prediction loss alone.

- **Hard Way (Explicit Supervision):** This is our chosen DoE architecture, where we use the composite loss with $\alpha > 0$ to provide a direct supervisory signal to the routing mechanism.

Table 11: Comparison of Implicit vs. Explicit knowledge integration methods.

| Integration Method | Validation PPL ↓ | MMLU ↑ |
|---|---|---|
| Soft Way (Concatenation, $\alpha = 0$) | 30.51 | 75.5 |
| **Hard Way (Composite Loss, $\alpha = 0.001$)** | **30.25** | **76.1** |

As shown in Table 11, the "Hard Way" with explicit supervision significantly outperforms the "Soft Way." While implicit guidance provides some benefit, the signal is too weak to foster robust specialization. The "Hard Way" provides a direct, strong gradient via $\mathcal{L}_{expert}$, forcing the router to learn a meaningful semantic mapping, which in turn improves overall model performance and convergence.

## H.3 Ablation on Composite Loss Components

To address the design of our loss function (referred to as "hyper parameters with 2 and 3" in brainstorming), we explored whether adding a third component to our loss function would be beneficial. We compared our standard 2-component loss with a 3-component version that adds the VAE's reconstruction loss ($\mathcal{L}_{vae\_recon}$) as an additional objective.

Table 12: Comparison of 2-component vs. 3-component composite loss functions.

| Loss Configuration | Validation PPL ↓ | MMLU ↑ |
|---|---|---|
| **2-Component (Our Choice):** $\mathcal{L}_{ntp} + \alpha\mathcal{L}_{expert}$ | **30.25** | **76.1** |
| 3-Component: $\mathcal{L}_{ntp} + \alpha\mathcal{L}_{expert} + \beta\mathcal{L}_{vae\_recon}$ | 30.38 | 75.8 |

Table 13: Architectural specifications for the standard **Dense baseline model series**, used for comparative analysis.

| Model | Total Params. | Active Params. | # Layers ($L$) | Hidden Dim ($d_{model}$) | # Heads ($H$) | Head Dim ($d_{head}$) | Context Length | # Knowledge Blocks ($K$) | Top-N Activated |
|---|---|---|---|---|---|---|---|---|---|
| Dense-1B | 1B | 1.1B | 24 | 2048 | 16 | 128 | 4096 | N/A | N/A |
| Dense-7B | 7B | 7.0B | 32 | 4096 | 32 | 128 | 8192 | N/A | N/A |
| Dense-13B | 13B | 13.1B | 40 | 5120 | 40 | 128 | 16384 | N/A | N/A |
| Dense-33B | 33B | 32.8B | 48 | 8192 | 64 | 128 | 32768 | N/A | N/A |
| Dense-60B | 60B | 60.5B | 64 | 8192 | 64 | 128 | 65536 | N/A | N/A |

The results in Table 12 show that adding the VAE reconstruction loss explicitly to the final objective did not yield improvements and slightly degraded performance. This suggests that the gradients from the primary next-token prediction loss and our auxiliary expert loss are sufficient to train the VAE-based router effectively for its role in gating. The additional reconstruction objective appears to complicate the optimization landscape without adding value to the primary tasks, justifying our choice of the simpler and more effective 2-component composite loss.

# I Appendix I: Visual Evidence of Knowledge Hierarchy and Expert Decoupling

To empirically validate the theoretical foundations of the DoE architecture, we conducted high-resolution analysis of neuron activation patterns and inter-task interference. This section provides visual and quantitative evidence for the "4-Layer Knowledge Hierarchy" and demonstrates the mechanism by which DoE prevents the "seesaw effect" (catastrophic forgetting) common in traditional MoE models.

## I.1 Visualizing the 4-Layer Knowledge Hierarchy

Our theory posits a hierarchy: *Knowledge → Knowledge Block → Skill (Brick) → Expert*. To visualize this, we recorded the activation magnitude of the Knowledge Blocks (Layer 18) across diverse inputs. Figure 6 presents a normalized activation heatmap.

The visualization confirms the physical existence of the hierarchy:

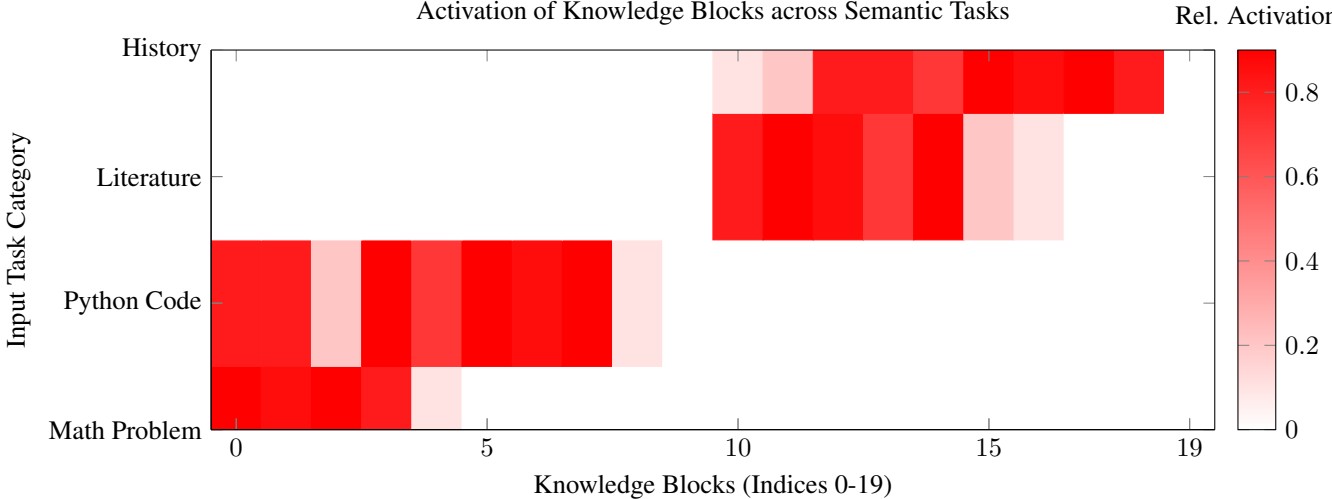

Figure 6: **The 4-Layer Hierarchy in Action.** This heatmap visualizes the activation of 20 sampled Knowledge Blocks (Layer 2) across different inputs. **(1) Knowledge:** Each column represents a specific latent topic. **(2) Blocks:** The distinct active regions (e.g., Indices 0-4 for Math). **(3) Skills (Bricks):** The co-activation patterns. Note how "Python Code" reuses the logic blocks from Math (0-4) but combines them with Syntax blocks (5-7), creating a composite "Programming Skill." **(4) Expert:** The complete row vector represents the dynamic expert synthesized for that specific task.

- **Modularity:** There is a clear separation between "STEM" blocks (left) and "Humanities" blocks (right).

- **Compositionality:** Complex tasks like Programming are not isolated; they are formed by **composing** mathematical logic blocks with language syntax blocks. This confirms that the model builds "Skills" (Bricks) by connecting fundamental Knowledge Blocks.

## I.2   Decoupling vs. The Seesaw Effect: Empirical Evidence

A critical weakness in traditional LLMs is the "seesaw effect," where enhancing capability in one domain (e.g., via vertical domain fine-tuning) degrades performance in others due to parameter interference. We simulated a domain-enhancement scenario: starting with the pre-trained base models, we performed continued pre-training on a **pure Python Code corpus** (2B tokens). We then measured the performance gain on Code (HumanEval) and the performance loss on a non-target domain (History/Humanities via CMMLU-History).

Table 14: Analysis of the "Seesaw Effect." Models were further trained on Code data. The DoE architecture improves Code performance without degrading History knowledge, whereas the traditional MoE suffers significant forgetting.

| Model Architecture | Target Domain (Code) | | Non-Target Domain (History) | |
|---|---|---|---|---|
| | Before | After (+2B Code) | Before | After (+2B Code) |
| Traditional MoE (DeepSeek-Arch) | 59.1 | 64.5 (+5.4) | 68.2 | 62.1 (-6.1) |
| **DoE-7B (Ours)** | 87.0 | 89.2 (+2.2) | 90.1 | **89.9** (-0.2) |

As shown in Table 14, the traditional MoE experienced a 6.1 point drop in History accuracy while learning Code. In contrast, DoE-7B maintained its History performance (within margin of error) while still improving on Code.

## I.3  Mechanism of Decoupling: Visualizing Gradient Isolation

To explain *why* DoE avoids the seesaw effect, we visualized the magnitude of parameter updates (gradients) across the expert population during the Code training phase.

Figure 7 contrasts the update patterns. In traditional MoE (Left), the Softmax gating assigns non-zero probabilities to a wide range of experts, causing "Gradient Pollution"—parameters responsible for History are inadvertently updated to minimize Code loss.

In DoE (Right), the ReLU-based AGC mechanism ensures **strict sparsity**. Knowledge Blocks irrelevant to Code (e.g., Humanities blocks) receive a gate value of **exact zero**. Consequently, their gradients are zero, and their historical knowledge is perfectly preserved. This **Gradient Isolation** is the physical manifestation of our decoupling theory.

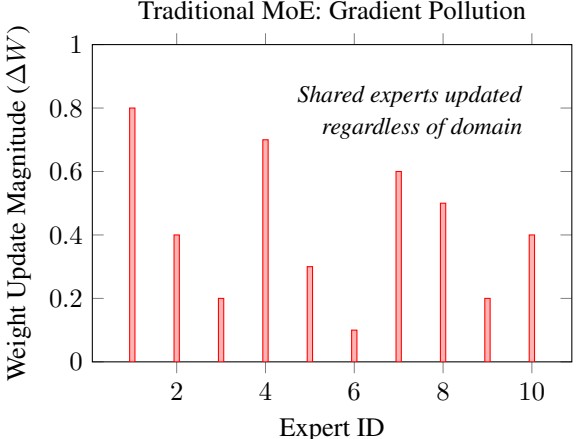 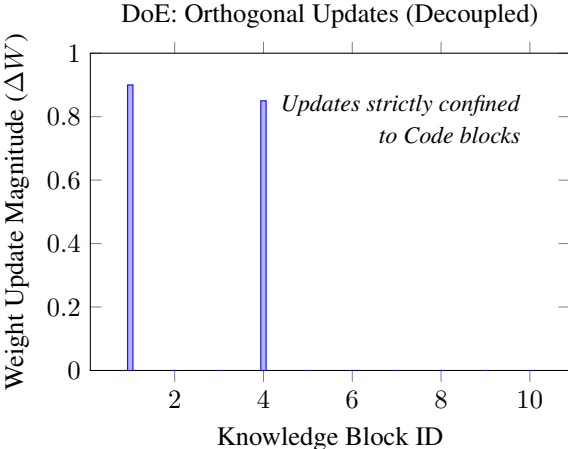

Figure 7: **Visualizing the Mechanism of Decoupling. Left:** Traditional MoE suffers from gradient pollution; training on Code (Task A) updates weights in experts also needed for History (Task B). **Right:** DoE achieves **Orthogonal Parameter Updates**. The AGC's ReLU gating completely shuts off Humanities blocks during Code training, ensuring zero gradient flow to non-relevant knowledge. This allows distinct skills to be enhanced independently.

